

# Global projections of aridity index for mid and long-term future based on CMIP6 scenarios

Camille Crapart[1], Sandrine Anquetin[1], Juliette Blanchet[1], Arona Diedhiou[1]

[1] Univ. Grenoble Alpes, IRD, CNRS, INRAE, Grenoble INP, IGE, 38000 Grenoble, France

*Correspondance to*: Camille Crapart (camille.crapart@univ-grenoble-alpes.fr)

**Abstract**

This study evaluates and projects global aridity index (AI) and dryland distribution using the FAO Aridity Index based on Penman-Monteith potential evapotranspiration. A multimodel ensemble of 13 CMIP6 models, with a horizontal resolution of 100 km, was selected for analysis. The ensemble was validated against WorldClim and ERA5 reanalysis datasets for the

reference period (1970–2000), showing strong correlations in key variables and consistent geographic representation of drylands, with some regional discrepancies, notably in North-Eastern Brazil. Future projections of AI were generated for three socio-economic pathways (SSP2-4.5, SSP3-7.0, and SSP5-8.5) and two timeframes (2030–2060 and 2070–2100). Results indicate that most regions will maintain their current climate classification but face decreasing AI values, signifying drier conditions. Under SSP2-4.5 and SSP5-8.5, significant drying is projected for the mid-term, with continued but slower changes

by century's end, affecting regions such as North and Central America, the Mediterranean Basin, and areas adjacent to present-day deserts. In contrast, SSP3-7.0 shows limited drying or localized wetting in the mid-term, followed by extensive drying in the long-term. Comprehensive maps and tables detailing dryland proportions and distributions are provided to support these findings.

## 1 Introduction

Ongoing climate changes raise concerns about the habitability of drylands, already facing challenges related to water availability, agriculture and population. Around 27% of the world inhabitants lived in drylands in 2020, i.e. a more than 2 billion people (Doxsey-Whitfield et al. 2015). Drylands are broadly defined as arid or semi-arid regions, i.e. regions in which the balance between water received and water loss is in favour of the latter. The concept of aridity refers to a long-term trend of limited water resources, contrarily to "drought" that refers to a temporary episode of water deficit. The IPCC defines aridity

as: "the state of a long-term climatic feature characterised by low average precipitation or available water in a region". Aridity generally arises from widespread persistent atmospheric subsidence or anticyclonic conditions, and from more localised subsidence on the lee side of mountains"(Möller, V. et al. 2022). Hyperarid and arid zones, such as the Sahara Desert, are mostly located at the descending side of Hadley cells. Semi-arid zones lie between the divergence zones of the two Hadley cells at the equator, and at the divergence zone of Hadley and Ferrel cells near the tropics of Cancer and Capricorn (Scholes



2020). Drylands are heterogenous and include various kinds of ecosystems, agricultural and economic activities. Their role in the global climate and biogeochemical cycles is still poorly understood (Osborne et al. 2022). On the contrary, "humid" areas have a water balance that tend to receive more water than they can use. These include tropical and temperature ecosystems, but also encompass great heterogeneity. In the recently released UNCDD report on desertification (Vincente-Serrano et al. 2024), it has been established that the three last decades saw 77.6% of the world getting dryer and that nearly 8% of the world

land surface transitioned to dryer aridity classes.

The classification of climatic zones based on the concept of aridity inherits from a long tradition of climate classification.

The Ancient Greeks used variables such as latitude and length of the longest day to divide the known world into torrid, temperate and frigid zones (Sanderson 1999). The Greek word "*klima*" means indeed "inclination of a sun ray" or latitude (Lamb et al. 2024). Maps of the world as the one drawn by Ptolemy of Alexandria were used until late in the Middle Ages

(Ptolémée 1561). In the 19[th] century, botanists and plant geographer defined better climatic zones based on the effect of temperature and precipitation on the plant types and distributions. One of the most widely used classification is the one from the botanist Wladimir Köppen who defined climatic zones based on several criteria, i.e. temperature, length of the winter months (Köppen**,** 1936, updated by Kottek et al. 2006 and Peel et al. 2007). In addition, many indices for climate classification have been introduced (Stephen 2005). For example, Lang defined a "rain factor" (Lang 1915), De Martonne an "aridity index"

(Martonne 1926), Emeberger a "pluviometric constant" (Emberger 1930) and Ångström a "coefficient of humidity" (Ångström 1936), all of them with their associated categories of climates.

Acknowledging the important contribution of Köppen to climate classification, Thorthwaite (Thornthwaite 1943) deplored the complexity of his classification and advocated for a physically-based, systematic, and concise way of differentiating the climates. He highlighted the importance of moisture and heat, and particularly of the processes of evapotranspiration.

Evapotranspiration is a complex processus to estimate, and in climate classifications, one uses the potential evapotranspiration i.e. the highest possible evapotranspiration given a good water supply (Xiang et al. 2020). In 1948, he introduced a moisture index that he uses for calculating the potential evapotranspiration (Thornthwaite 1948). In parallel, Penman (Penman 1948) derived his evapotranspiration equation from the surface energy balance. Monteith (Monteith 1965) built on this work to establish the Penman-Monteith (PM) equation recognized as the most complete way of calculating evapotranspiration.

However, its extensive need in terms of variables makes other and simpler equations also widely used (Pimentel et al. 2023). For example, later work by Hargreaves et Allen (2003) provided a simpler method with the aim of guiding irrigation practices in arid and semi-arid zones.

The three successive editions of the World Atlas of Desertification (United Nations Environment Programme 1992; Nick Middleton et David Thomas 1997; Joint Research Centre (European Commission) et al. 2018) provided maps of aridity zones,

using the Thornthwaite equation for its simplicity. In the 2024 UNCCD report, the Hargreaves question is used for calculating evapotranspiration (Vincente-Serrano et al. 2024). Some other authors used the Penman-Monteith index on the reference period 1970-2000 to provide world maps, such as the FAO (FAO 2021) and Zomer et al. (2022). Spinoni et al. (2015) identified regions prone to desertification by comparing the 1951-1980 and 1981-2010 aridity indexes calculated with the PM potential



evapotranspiration. By comparing with the literature, they showed that the regions identified as at risk are areas where
desertification or land degradation is reported. Other studies of climate zones by the end of the century are available, such as
a Köppen classification until 2100 with CMIP6 was done by Beck et al. (2023). These maps are very detailed, but do not
provide information on the changes within each climate category. Similarly, Trabucco et al. 2024 also published global maps
of aridity index for the periods 2021-2040 and 2041-2060, using the downscaled models available in Worldclim (Fick and
Hijmans 2017). Due to the few variables available in the future downscaled CMIP6 models gathered in Worldclim, the authors
had to use the Hargreaves equation for calculating the reference potential evapotranspiration. In addition, these maps of future
aridity areas are not available for the end of the century, and the pertinence of CMIP6 models is not evaluated. Using
temperature-based methods like Hargreaves or Thornthwaite methods tend to overestimate the potential evapotranspiration in
the long-term, by ignoring the effects of wind, radiation and shading (Sheffield, Wood, et Roderick 2012). The Penman-
Monteith method includes these factors and is less reliant on temperature. In general, no future estimations of the aridity index
globally, mid-term and long-term, calculated with the Penman-Monteith reference potential evapotranspiration is available.

In this study, we intend to compute the global aridity index based on Penman-Monteith equation globally for two periods: mid-
term (2030-2060) and long-term (2070-2100), using CMIP6 models. This allows us to identify the areas prone to aridification
in the short and long term, including within areas defined as "humid", and provide maps of aridity category areas for three
socio-economic pathways (SSP 2-4.5, SSP 3-7.0 and SSP 5-8.5). In a first part, the performance of the CMIP6 ensemble is
evaluated for the reference period 1970-2000 by the comparison with two databases. The first database is the widely used
Worldclim which is a combination of observations and reanalysis, and provides the 30 years average of several bioclimatic
variables. The second one is the ERA5 reanalysis. ERA5 and Worldclim had very similar patterns of precipitation and
temperature and were considered equally good as references. In the second part of the article, we compare the evolution of
aridity index in each grid cell in three Socio-Economic Pathways (SSP) between the reference period 1970-2000 and the two
study periods, 2030-2060 and 2070-2100. We compare the change in aridity index with the projected changes in temperature
and precipitation, disentangling the relative role of these two factors in climate change. Finally, we examine the areas that will
exceed the threshold separating aridity categories, and provide a map of aridity categories in each scenario.



## 2 Material and methods

### 2.1 Aridity index

The aridity index used in this study was first introduced by the UNESCO in 1979 to establish a world map of drylands prior to the United Nations Conference on Desertification (UNESCO 1979). It uses the Penman-Monteith equation to calculate the potential evapotranspiration, with standardized parameters adapted to an area of growing crops and noted $ET_0$ (Allen et al. 1998). This equation is an adaptation of the energy balance at the surface to calculate the quantity of water lost through

evapotranspiration under optimum irrigation conditions, in mm per day. The aridity index is the average annual precipitation over 30 years, divided by the average annual potential evapotranspiration over 30 years, expressed in the World Atlas of Desertification (Joint Research Centre (European Commission) et al. 2018) as:

$$AI = \frac{\sum_{i=1}^{30} \frac{P_i}{ET_{0_i}}}{30} \tag{1}$$

The Penman Monteith equation for the potential evapotranspiration is:

$$ET_0 = \frac{0.408\Delta(R_n - G) + \gamma \frac{900}{T + 273} u_2(e_s - e_a)}{\Delta + \gamma(1 + 0.34u_2)} \tag{2}$$

Where $ET_0$ is the monthly potential evapotranspiration in mm, $R_n$ is the net surface radiation in MJ/m²/day, G is the soil heat

flux density in mJ/m²/day, T is the mean daily temperature at 2m height in °C, $u_2$ is the wind speed at 2m height in m/s, $e_s$ and $e_a$ are the saturating and actual vapour pressure in kPA. $\Delta$ is the slope of the vapour pressure curve in kPa/°C and $\gamma$ is the psychrometric constant in kPa/°C, that depends on atmospheric pressure and temperature. This equation is an adaptation of the general equation of evapotranspiration from Penman-Monteith for a hypothetical surface planted with crops and used to homogenise the parameters related to the vegetation.

Annual precipitation was obtained by adding mean monthly values. Similarly, annual $ET_0$, was obtained by calculating the mean $ET_0$ for each month over 30 years and then summed the monthly values. This was preferred over averaging the monthly values of all variables (temperature, wind speed, radiation....) for use in equation (2), due to the non-linearity of the Penman-Monteith formula.

This would be represented by:

$$ET_0 = \sum_{j=1}^{12} \frac{\sum_{i=1}^{30} ET_{0_{i,j}}}{30} \tag{3}$$




Where "j" represents the months of the year and 30 the years on which the aridity index is calculated. $ET_{0_{i,j}}$ is the potential evaporation in mm for a given month in a given year "i".

The climate is then classified into 5 classes depending on their aridity index. The explicative note of the UNESCO (UNESCO 1979) on the map of the world's arid regions gives more detail on the vegetation present on these zones:


**Table 1 - Categories of aridity in the UNESCO classification**

| | |
|---|---|
| AI < 0.03 | Hyperarid <br> Desert, no perennial vegetation. |
| 0.03 < AI < 0.2 | Arid <br> Scattered vegetation like bushes and shrubs |
| 0.2 < AI < 0.5 | Semi-arid <br> Savannah, sometimes grazing/agriculture areas |
| 0.5 < AI < 0.75 | Dry subhumid <br> Savannah, maquis, chaparral. |
| AI > 0.75 | Humid |
| $ET_0$ < 400 mm | Cold |

In this note, there is no mention of cold regions (Northern Europe, Siberia, Greenland). However, in the World Atlas of Desertification (Joint Research Centre (European Commission) et al. 2018) a "cold" region is defined, in which the annual potential evapotranspiration is inferior to 400 mm/year. In our data, grid cells with $ET_0$ inferior to 400 mm have either an aridity index classified as humid, either the annual evapotranspiration is calculated as negative and the index is also negative. To avoid this last case, we decided to integrate the "cold" category, defined as grid cells in which $ET_0$ is inferior to 400 mm/year. In this classification, "drylands" comprise all the categories outside "humid" and "cold".

## 2.2 Variables and climate databases

All data analysis was performed using R programming software (R Core Team 2023). Data from different models were reprojected into the same grid, and extracted using the «raster» package (Hijmans et al. 2023).

The land-sea mask used is extracted from Iturbide et al. (2020), which also provides the polygons of the regions defined for the 6th Assessment Report.

Elevation data (used to calculate atmospheric pressure) are extracted from Worldclim and used for the 3 datasets.

### 2.2.1 Source of data

Historical climate data are taken from the Worldclim database (Fick and Hijmans 2017) and from the ERA5 monthly aggregated reanalysis (Hersbach et al. 2020).

Worldclim is composed of a combination of observations and reanalysis averaged monthly over a 30 years period (1970-2000). We used the coarser resolution: 340 km².



ERA5 offers reanalysis of atmospheric variables monthly aggregated with a horizontal spatial resolution of 31 km. The data were downloaded for the period 1970-2000 and averaged per grid cell and by month over 30 years.

CMIP6 models are accessible through the Lawrence Livermore National Laboratory, one of the ESGF data nodes (« LLNL ESGF MetaGrid »). We filtered the available models based on the following criteria:

- The horizontal spatial resolution is 100 km;

- The 6 necessary variables are available: air temperature at 2m height (« tas »), precipitation (« pr »), surface wind speed at 2m height (« sfcWind »), surface latent heat flux (« hfls »), surface sensible heat flux (« hfss »), relative humidity (« hurs »).

- These variables are simulated for the 4 following scenarios: historical (years 1850-2014), SSP 2-4.5, SSP 3-7.0 and SSP 5-8.5 (years 2015-2100).

13 models were thus selected are:

- CAS-ESM2-0 (Zhang et al. 2020) ;
- CESM2-WACC (Gettelman et al. 2019);
- CMCC-CM2-SR5 (Cherchi et al. 2019);
- CMCC-ESM2 (Lovato et al. 2022)
- CNRM-CM6-1 (Voldoire et al. 2019);
- EC-Earth3 (Döscher et al. 2022);
- FGOALS-f3-L (B. He et al. 2019);
- GFDL-ESM4 (Dunne et al. 2020);
- INM-CM4-8 and INM-CM5-0 (Volodin et al. 2018);
- MPI-ESM1-2 (Gutjahr et al. 2019);
- MRI-ESM2-0 (Yukimoto et al. 2019);
- NorESM2-MM (Seland et al. 2020).

Only one member of each of them was downloaded, usually r1i1p1f1 except for the CNRM model which only provided the member r1i1p1f2.

For each cell in the grid, the "CMIP6" value is the multimodel mean value of a given variable. The standard deviation was computed to estimate the spread of the models.

**2.2.2 Regions and land/ocean mask**

The land/ocean mask, the continent and the corresponding IPCC regions were obtained from Iturbide et al. (2020). Grid cells containing only ocean (marked by a value of 0) were excluded, but the coastal grid cells (value between 0 and 1) were kept in

the analysis. We excluded from the analysis the continents that were only composed of a few grid cells, mainly islands, i.e. the




"Arctic", "Indian', "Pacific", "Polar" and "Southern" continents. In addition, most figures and percentages in the article exclude the grid cells of Antarctica, unless otherwise specified.

### 2.2.3 Variables

The potential evapotranspiration is calculated based on the variables available in each database. Table 2 summarises which
variable is used to compute each term in the Penman-Monteith equation.

**Table 2 – List of variables used by data source**

| Calculated variable | Worldclim | ERA5 | CMIP6 |
|---|---|---|---|
| Annual precipitation | "precip" in mm/y | "mpr" in kg/m2/s Monthly aggregated, averaged over a year | "pr" in kg/m2/s Monthly aggregated, averaged over a year |
| 2m temperature | "tavg" in °C Available as the mean of average monthly temperature for 30 years (1970-2000) | "t2m" in K Used to calculate evapotranspiration by month | "tas" in K Used to calculate evapotranspiration by month |
| 2m wind speed | "wind" in m/s Wind speed at 10 m. Converted to wind speed at 2m by multiplying by 0.748 | "si10" in m/s Wind speed at 10 m. Converted to wind speed at 2m by multiplying by 0.748 | "sfcWind" in m/s Wind speed at 10 m. Converted to wind speed at 2m by multiplying by 0.748 |
| Net solar radiation and soil heat flux Rn – G | Rn is estimated from the solar radiation "srad" in kJ/m2/day. G is neglected | Rn – G is computed as the sum of the surface latent heat flux "mslhf" and the surface sensible heat flux "msshf" | Rn – G is computed as the sum of the surface latent heat flux "hfls" and the surface sensible heat flux "hfss" |
| Saturating vapor pressure $e_s$ | Calculated from 2m temperature | Calculated from 2m temperature | Calculated from 2m temperature |
| Actual vapor pressure $e_a$ | Directly available as water vapor pressure "vapr" in kPa | Calculated from the dew point at 2m "tdew" in K | Calculated from the relative humidity "hurs" in % |

### 2.2.4 Computation of aridity indexes and categories

We use the evapotranspiration as defined by Penman-Monteith, that requires the mean annual temperature, actual vapor pressure and surface energy fluxes. 13 CMIP6 models were selected, that offered the 6 necessary variables were readily
available for a resolution of 100 km and for the historical period as well as the 3 SSP.



We retrieved data for future periods in CMIP6 in 3 distinct SSP scenarios. The potential evapotranspiration and the aridity index are then calculated for these 3 scenarios in the 13 CMIP6 models. The average aridity index is computed by grid cells and the value of this index in the two future periods (2030-2060 and 2070-2100) are computed. Increases in aridity index, corresponding to wetter conditions, are represented in blue; while decreases of aridity index, corresponding to dryer conditions,

are represented in red. Most of the values taken by the aridity index are comprised between 0 and 1, corresponding to the arid, semi-arid and dry-subhumid categories. Aridity indexes superior to 0.65 up to infinity are classified as humid, except grid cells with evapotranspiration lower than 400 mm/year that are classified as cold. Aridity indexes less than 0.03 to (-infinity) are classified as hyperarid. Figures for changes in temperature (the main driver for evapotranspiration) and precipitation compared to 1970-2000 are available in supplementary (Fig. S5).

**3 Evaluation of CMIP6 performances for the reference period 1970-2000**

**3.1 Internal variability in CMIP6**

Before calculating the multimodel AI average, the aridity index and corresponding aridity category are calculated for each model in each period and for each grid cell. Then an aridity category is assigned to the grid cell based on the calculated value (Table 1). A summary of the global percentage of aridity category by model is presented in Table 3, excluding Antarctica.

The multimodel mean is computed for each grid cell, and the aridity category is determined by this multimodel value. Hence, the multimodel mean in percent is not equal to the mean of the percentages for the 13 models. The percentages of hyperarid, arid, semi-arid and dry-subhumid grid cells are merged into a category "Drylands". In the 13 models used in this study, this percentage of drylands varies from 24.5% to 38%. Only 3 models have a percentage of drylands inferior to the multimodel average. Two of the models had a particularly high proportion of "Cold" grid cells (ET0 < 400 mm/year) compared to the

"Humid" category: CMCC-ESM2 and FGOALS.

Unsurprisingly, models developed by the same institution are very similar. CMCC-CM2-SR5 and CMCC-ESM2 are 2 of the 3 "wettest" models, and INM-CM4-8 and INM-CM5-0 also have close results in terms of proportion of drylands. Some of the models in our subset have similarities in the code and results (Pathak et al. 2023). However, this does not always result in

similar proportion of aridity categories. For example, EC-Earth-3 and CNRM-CM6-1-HR are supposed to be correlated, and share parts of their code. They have a similar proportion of drylands here (35.8% and 33.1%), but differ in their proportion of "Cold" areas (25.7% compared to 31%). Similarly, NorESM2, which is supposed to be similar to CESM2-WACCM and CMCC models, has the highest proportion of drylands (38.0%), while the two CMCC models have the lowest one. Given this variability even in models supposed to be similar, we chose not to weight the models for the multimodel mean.

As a result, the multimodel average is strongly influenced by the wettest and coldest models, with a total proportion of drylands of 28.3%. However, the multimodel geographical repartition of aridity areas is more consistent with observations than the repartition in each individual model (Fig. S1). The multimodel average is also consistent with the two reference databases, as





demonstrated below. Internal variability will not be further investigated, and we will use the multimodel average from now on.


**Table 3 – Percentage of aridity categories for the 13 CMIP6 models and multimodel average percentage of aridity categories for the reference period 1970-2000. The multimodel average and corresponding standard deviation from now on refer to the global average, and not the average per grid cell.**

| Model | Hyper-arid | Arid | Semi-arid | Dry subhumid | Sum drylands | Humid | Cold | NA |
|---|---|---|---|---|---|---|---|---|
| CAS-ESM2-0 | 5.2 | 8.6 | 13.2 | 6.3 | 33.3 | 37.4 | 29.1 | 0.3 |
| CESM2-WACCM | 5.3 | 10.7 | 14.1 | 5.2 | 35.3 | 35.0 | 29.5 | 0.3 |
| CMCC-CM2-SR5 | 5.1 | 7.4 | 10.1 | 5.4 | 28.0 | 42.2 | 29.5 | 0.3 |
| CMCC-ESM2 | 6.4 | 7.9 | 7.7 | 2.5 | 24.5 | 24.4 | 50.8 | 0.3 |
| CNRM-CM6-1 | 5.8 | 12.2 | 11.3 | 6.5 | 35.8 | 38.3 | 25.7 | 0.2 |
| EC-Earth3 | 8.4 | 9.5 | 10.8 | 4.4 | 33.1 | 35.7 | 31.0 | 0.2 |
| FGOALS-f3-L | 5.9 | 10.9 | 10.0 | 4.9 | 31.7 | 25.1 | 42.9 | 0.2 |
| GFDL-ESM4 | 5.8 | 8.9 | 8.9 | 4.1 | 27.7 | 35.3 | 36.8 | 0.2 |
| INM-CM4-8 | 4.4 | 8.4 | 13.3 | 5.7 | 31.8 | 40.1 | 27.8 | 0.3 |
| INM-CM5-0 | 3.4 | 8.5 | 12.0 | 5.6 | 29.5 | 43.0 | 27.3 | 0.3 |
| MPI-ESM1-2 | 8.7 | 11.1 | 10.2 | 4.3 | 34.3 | 34.0 | 31.5 | 0.2 |
| MRI-ESM2-0 | 6.1 | 10.8 | 9.2 | 3.9 | 30.0 | 38.9 | 30.9 | 0.3 |
| NorESM-2-MM | 5.2 | 10.1 | 17.1 | 5.6 | 38.0 | 33.4 | 28.3 | 0.3 |
| **Multimodel average** | **5.8** | **9.6** | **11.4** | **5.0** | **31.8** | **35.6** | **32.4** | **0.3** |
| **Standard Deviation** | **1.4** | **1.5** | **2.5** | **1.1** | **6.5** | **5.7** | **7.1** | **0.1** |





## 3.2 Comparison between Worldclim, ERA5 and multimodel CMIP6

The Worldclim database is commonly used in disciplines such as ecology, biology or biogeochemistry, whereas climatologists rather use more detailed ensembles such as ERA5. These two databases were compared as a basis for evaluating the historical models of the CMIP6 ensemble. Figure 1 shows the violin plots of the main variables in CMIP6, ERA5 and Worldclim: annual mean precipitation, surface temperature, surface wind speed, solar radiation, actual vapor pressure, and the computed potential evapotranspiration. In addition, Table 4 shows the r2 of these variables between CMIP6, ERA5 and Worldclim.


Figure 1 – Violin plots of the main climate variables in Worldclim, ERA5 and CMIP6. Rn-G represents the net solar radiation minus the soil heat flux, and represents the total heat fluxes at the surface. "ea" is the actual vapor pressure.

All the variables are well correlated with each other, up to an $r^2$ of 1 for surface temperature for the 3 pairs of databases. The actual vapor pressure in the three databases is also highly correlated (r = 1 for ERA5/Worldclim, and 0.99 for ERA5/CMIP6





and Worldclim/CMIP6). The annual precipitation, the wind speed and the solar radiation have a higher spread between databases. The shapes of the violins for precipitation are similar, despite higher values in CMIP6. The correlation coefficients are close to 0.9 (r = 0.88 for ERA5/Worldclim, r = 0.89 for ERA5/CMIP6 and Worldclim/CMIP6). The spread is largest for

the wind speed, where ERA5 and CMIP6 are more correlated (r = 0.88 for ERA5/CMIP6, r = 0.77 for ERA5/Worldclim and r = 0.78 for Worldclim/CMIP6). However, the values of wind speed in CMIP6 are mostly comprised between 3 and 5 m/s, while the range is broader in ERA5 and Worldclim. The solar radiation (Rn-G) is also more correlated between ERA5 and CMIP6 than compared to Worldclim (r = 0.99 for ERA5/CMIP6, while r = 0.89 for ERA5/Worldclim and 0.91 for Worldclim/CMIP6). This is also reflected in the shape of the density, with Worldclim having more negative values than ERA5

and CMIP6, no points higher than 12 MJ/m2/day, and most values being comprised between 5 and 10 MJ/m$^2$/day. This difference is due to the different source for the two variables: in CMIP6 and ERA5, Rn – G is calculated as the sum of the flux of latent and sensible heat fluxes, while in Worldclim this term is derived from the total solar radiation and the latitude.

Overall, the discrepancy in wind speed and solar radiation does not impact the strong correlation between the calculated ET0 in the three databases: r = 0.97 for ERA5/Worldclim, 0.98 for ERA5/CMIP6, and 0.96 for Worldclim/CMIP6. The differences

are more reflected in the median value: ET0 in CMIP6 (809,7 mm/year) is similar to that in ERA5 (809,2 mm/year), but ET0 have lower values in Worldclim (median ET0 = 775,2 mm/year).

**Table 4- r$^2$ for variables, pairwise comparison of databases. All p-values were <2×10$^{-16}$**

|  |  | Precipitation | Temperature | Wind speed | Rn - G | Actual vapor pressure (ea) | Potential evapotranspiration (ET0) |
|---|---|---|---|---|---|---|---|
| ERA5 vs Worldclim |  |  |  |  |  |  |  |
|  | r$^2$ | 0.77 | 1.00 | 0.59 | 0.79 | 0.99 | 0.94 |
| ERA5 vs CMIP6 |  |  |  |  |  |  |  |
|  | r$^2$ | 0.79 | 1.00 | 0.77 | 0.97 | 0.98 | 0.96 |
| Worldclim vs CMIP6 |  |  |  |  |  |  |  |
|  | r$^2$ | 0.79 | 0.99 | 0.61 | 0.82 | 0.97 | 0.91 |

The 30-years average of the aridity index is used to compare ERA5, Worldclim and CMIP6 datasets for the reference period 1970-2000. Figure 2 presents a pie chart showing the percentage of each aridity category for the 3 datasets and for the reference period.



The pie charts highlight the similarities and differences between CMIP6 and the two reference databases. In general, there are

less grid cells classified as drylands in CMIP6 (28.3%) compared to ERA5 (30.4%) and Worldclim (32.3%). The proportion of humid grid cells is the highest in ERA5 (42.4%) compared to CMIP6 (40.1%) and Worldclim (37.3%). CMIP6 has the highest share of cold grid cells (31.6%). This share decreases to 27.1 % in ERA5 and 30.4% in Worldclim.

Regional differences in the distribution of aridity categories are visible on Figure 3. Overall, the CMIP6 multimodel categories matches the patterns found in ERA5 and Worldclim. However, several deserts appear in CMIP6 as semi-arid or even dry

subhumid areas, whereas they are clearly arid or hyperarid in Worldclim and ERA5.

*North America:* deserts are less widespread in CMIP6 compared to the 2 others databases. For example, the Chihuahuan Desert, at the frontier between Mexico and the United States, is in a region that appears as semi-arid or even subhumid in CMIP6. The Great Basin Desert is also much smaller in CMIP6 than in ERA5 and Worldclim. CMIP6 ensemble has proven to have a wet bias in particular over Western US compared to observations (Almazroui et al. 2021).

*Central America:* CMIP6 is slightly dryer than ERA5 and Worldclim, with some dry subhumid and semi-arid grid cells in the Yucatan Peninsula, Cuba, Haiti and the north of Venezuela. These areas are classified as tropical savannahs in the Köppen-Geiger classification, with a dry winter season (Kottek et al. 2006). This dryer classification can be explained by the dry bias identified by Almazroui et al. (2021) in south Central America and the Caribbean.

*South America:* In South America, the main differences are visible in the North of Chile (Atacama Desert) and in North Eastern

Brazil. North Eastern Brazil is semi-arid in Worldclim and ERA5, but is completely humid in the multimodel CMIP6. Similarly, the Atacama Desert is not hyperarid in CMIP6: only an "arid" band appears. This wet bias has been observed earlier, for example by Reboita et al. (2024) who compared CMIP6 ensemble with reanalysis of temperature (ERA5) and precipitation (Climate Prediction Center Merged Analysis of Precipitation CMAP and Global Precipitation Climatology Project) in 5 subregions of south America. They observed a systematic wet bias in North-eastern Brazil in summer, as well as in the Andean

region. (Firpo et al. 2022) highlight the dipolar bias in precipitation in North-eastern Brazil and in the Amazonas, by comparing multimodel CMIP6 to the CRU database (Harris et al. 2014). They explain this by a default in the modelling of the maximum precipitation centre in South America, which is located too far east. One of the reasons could be a poor representation of cloud physics. This deficiency was already present in CMIP3 and 5. A bias towards warmer temperatures in south of South America was also found, but this does not seem to influence the distribution of aridity zones.

*Europe:* The European continent is divided into humid and cold zones, which do not differ between CMIP6 and ERA5/Worldclim.

*Mediterranean Basin:* CMIP6 multimodel ensemble and ERA and Worldclim differ in the Iberian Peninsula. The center and the east of the peninsula are semi-arid and dry subhumid in Worldclim and ERA5, while semi-arid areas are limited to the south in CMIP6. The same pattern is observed in Turkey. It is mostly humid in CMIP6, but the central plateau of Anatolia is

semi-arid or dry-subhumid in ERA5 and Worldclim, which corresponds more to the continental conditions observed.

*Africa*: CMIP6 performs well with all zones well represented. The arid zone in south-west Africa (Namibia, South Africa) is less spread in CMIP6 than in ERA5 and Worldclim. This is consistent with the evaluation made by Almazroui et al. (2020),





who noticed a wet bias in this region when comparing CMIP6 to the observations from the Climate Research Unit (University of East Anglia, Harris et al. 2014). In the Arabic peninsula, ERA5 has more hyperarid zones than CMIP6 and Worldclim, but

the entire peninsula is arid. This is consistent with reality.

*Asia:* In Western Central Asia and in India, arid and semi-arid zones are well represented in the three datasets of the three databases. Iran, Turkmenistan (70% covered by the Kara-Kum cold desert), Uzbekistan and Kazakhstan are mostly arid and semi-arid. However, there are differences in the Tibetan Plateau. In Worldclim, it is classified as arid or semi-arid, whereas some parts are classified as "Cold" in ERA5 and CMIP6. This is due to the particular way "Cold" areas are classified, based

on an annual potential evapotranspiration inferior to 400 mm/day. A cold bias has been found in the region resulting in an underestimation of the potential evapotranspiration (Zhu et Yang 2020). In particular temperature in the winter is much colder than observations in a majority of models. This impacts the evapotranspiration: more "cold" grid cells in CMIP6 compared to ERA5. The Gobi and Taklamakan deserts in Western China are visible in the three datasets, but only ERA5 identifies hyperarid areas.

*Oceania:* the Australian deserts appears as mostly semi-arid in CMIP6, while it is mostly arid in ERA5 and Worldclim. Only the Great Victoria Desert and its closest neighbours are arid, while the Great Sandy Desert, in North Western Australia, is only semi-arid.

Individually, some of the models have a stronger signal in the areas that the CMIP6 multimodel ensemble does not classify as drylands. For example, CESM and NorESM distinctly show North-Eastern Brazil as semi-arid. In addition, the Gobi and

Taklamakan deserts are identified as hyperarids in CAS-ESM2, CNRM, EC-Earth-3, MPI, and the deserts of Australia are better represented in the CNRM, EC-Earth3, FGOALS, GFDL-ESM4, and MRI models. Maps are provided in the Supplementary (Fig. S1).

However, globally, the multimodel average performs better than any individual model. Table 5 shows the percent of grid cells classified into different aridity categories for the 13 CMIP6 models compared to Worldclim and ERA5, as well as the

multimodel average. The percentage of difference between Worldclim and ERA5 is 13.1%. This percentage rises to 14.7 % when comparing the multimodel average to ERA5 and to 15.1% when comparing with Worldclim, which is better than any of the CMIP6 models taken individually.

To conclude, the multimodel average reproduces correctly the aridity areas corresponding the observations and reanalysis of Worldclim and ERA5.




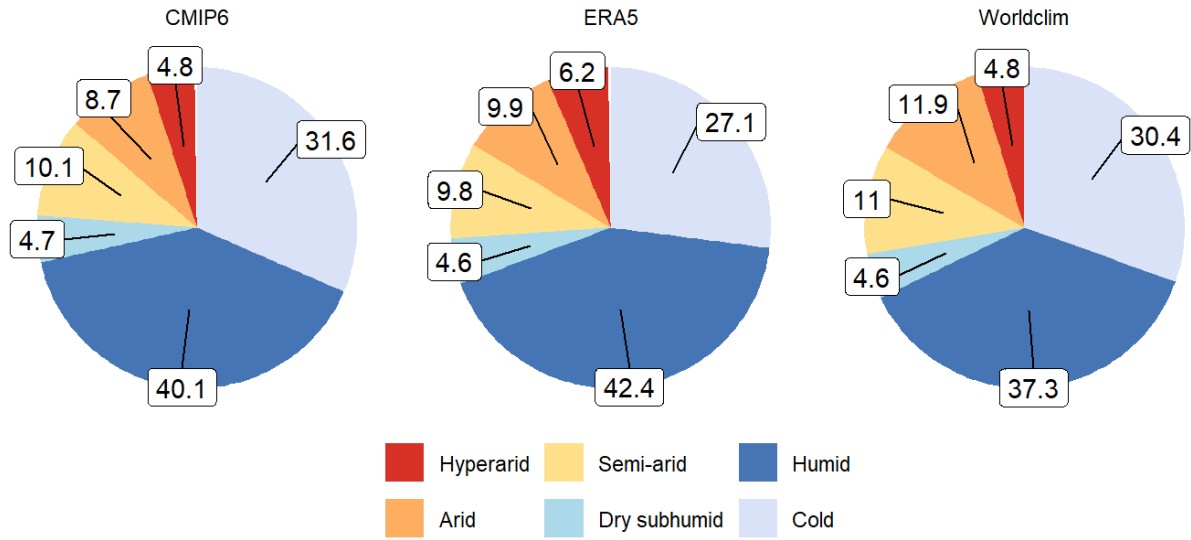

**Figure 2 – Pie chart of the proportion of aridity categories for the datasets CMIP6, ERA5, and Worldclim, excluding Antarctica**





**Table 5 - Proportion of gridcells with a different aridity category for CMIP6 models and for the multimodel average, compared to ERA5 and Worldclim**

| Model | Gridcell different from ERA 5, in % | Gridcells different from Worldclim, in % |
|---|---|---|
| **CAS-ESM2-0** | 18.1 | 19.7 |
| **CESM2-WACCM** | 16.4 | 15.9 |
| **CMCC-CM2-SR5** | 17.6 | 18.5 |
| **CMCC-ESM2** | 32.1 | 30 |
| **CNRM-CM6-1** | 17.8 | 21.2 |
| **EC-Earth3** | 16.7 | 17 |
| **FGOALS-f3-L** | 24.9 | 23.4 |
| **GFDL-ESM4** | 17.8 | 16.3 |
| **INM-CM4-8** | 19.3 | 19.9 |
| **INM-CM5-0** | 18.5 | 19.9 |
| **MPI-ESM1-2** | 16.7 | 19.7 |
| **MRI-ESM2-0** | 13.4 | 15.4 |
| **NorESM-2-MM** | 17 | 17.7 |
| **Multimodel average** | **14.7** | **15.1** |



Wordclim, 1970-2000

ERA5, 1970-2000

CMIP6, 1970-2000

Hyperarid   Semi-arid   Humid
Arid        Dry subhumid   Cold

**Figure 3 - Maps of aridity categories for the reference period 1970-2000 for Worldclim, ERA5 and multimodel CMIP6. The maps do not show the Antarctic continent (entirely "cold"), but Antarctica is included in the computation of the proportion of aridity categories.**





## 4 Future evolution of the aridity index in CMIP6

Here, we intend to diagnose the regions in which climate changes in term of aridity are the most susceptible to happen. We project aridity zones using 13 CMIP6 climate models for 3 of the socio-economic trajectories described in the IPCC's 6th Assessment Report: SSP 2-4.5, SSP 3-7.0 and SSP 5-8.5, for 3 past periods (1850-1880, 1970-2000 and 1985-2015) and 2 future periods (2030-2060 and 2070-2100). The SSP-4.5 corresponds to a "middle-of-the-road" scenario in which the emissions remain around current level until 2030, after which most countries acheive their net-zero targets for 2050 under the Paris Agreement. The SSP 3-7.0 is a "regional rivalry" scenario, in which each region acts for itself. No additional climate policy is taken by 2100, and emissions double compared to current levels. In this scenario, emissions include particularly high levels of non-CO2 greenhouse gases and the highest levels of aerosol emissions. The SSP 5-8.5 is also a scenario without any additional climate policy and where future economic development is based on an intensive use of fossil fuels (Chen et al. 2021).

### 4.1 Evolution of the aridity index value

Figure 4 presents the difference of aridity index between future periods (2030-2070 and 2070-2100) and the reference period (1970-2000) for the three studied SSP. The difference is presented in % for a better understanding, i.e. $AI_{(2070-2100)} - AI_{(1970-2000)} / AI_{(1970-2000)}$.

*Polar region*: The most impressive changes are located in Greenland, in the northernmost regions of North America, and in the polar archipelagos of Svalbard, Novaya Zemlya and Svernaya Zemlya. The changes vary greatly from cell to cell, ranging from -40 to + 40 %. This is probably due to the large changes in precipitation and temperature in this region, and to the difficulty of modelling the polar regions. The East coast of Greenland is less affected, with a decrease in the aridity index of about -20%. The temperature increases reach 4 to 5 °C in SSP2-4.5, but up to 8-10 °C in SSP5-8.5. In the latter scenario, Greenland experiences a temperature increase of 4 to 9°C, from south to north. In contrast, precipitation is projected to increase in all these areas, especially in the eastern part of Greenland (+20% in SSP2-4.5 and +40% in SSP5-8.5).

*North America:* The strongest aridity changes in North America occur mainly in Alaska, with patterns reminding Greenland, and around the current desertic regions. In the SSP2-4.5, the whole of Mexico and most of the south of the USA experience a 20% drying by 2030-2060, increasing to 30% drying in Mexico and in the South USA by 2070-2100. In particular, the Chihuahuan and Sonoran deserts, which straddle the southern USA and Mexico, become increasingly dryer with changes in aridity down to -40% in the SSP5-8.5. The changes in precipitation are relatively similar in the two SSP (-10 to -20%) but the temperature increase is more drastic in SSP5-8.5 compared to SSP2-4.5: +2.5 °C on average in Mexico in SSP5-8.5 compared to +1.5 °C in Mexico and South USA in SSP2-4.5. The rest of Northern America also warms up by an average of 3 °C.







**Figure 4 - Change in % of aridity index compared to the reference period 1970-2000 for SSP2-4.5, 370 and 585 for two future periods: 2030-2060 and 2070-210. Hatched areas correspond to areas where at least 10 models over 13 agree on the sign of change.**

*Central and South America:* In all scenarios, a decrease at least equal to 20% of the aridity index is observed in Central

America, on the Caribbean coast, in the central Brazil and in the southern part of Chile and Argentina, with the highest decrease





observed for the SSP5-8.5, period 2070-2100. The north-east part of Brazil, nowadays classified as a desert, is located in a region where there is almost no change in aridity index. Overall, the temperature increase in South America increases, from 1 to 3 °C in SSP2-4.5 to more than 4°C in SSP5-8.5 In SSP3-7.0, the increase in temperature is comprised between 2 and 4 °C.

In the three SSP, the precipitation patterns change mainly in the south of the continent (precipitation decreases in the south of Chile and increases in some parts of Patagonia). In SSP5-8.5, there is a general decrease of precipitation in Central America. These decreases coincide with the areas where the strongest changes of aridity index are predicted.

*Europe and Mediterranean basin*: In SSP2-4.5, there is overall decrease in the aridity index of between -10 and -20%. This excludes the Mediterranean coast, especially Spain, Italy, Greece, South-west France, and Ukraine. The aridity index decreases

by more than -30% in the South of Spain. In SSP5-8.5, the same pattern with more regions with decreases below -30%: whole of Spain and Italy, Yugoslavia, Greece etc. The region of up to 20% drying extends to the North of France and most of Germany and Poland. Parts of Northern Norway and Sweden are affected by similarly large decreases in aridity index, as well as the high plateaus of western Norway, and Iceland. In most of Europe, this pattern results from an overall higher temperature, which reaches up to +3-4 °C in SSP2-4.5 in continental Europe and up to 5 to 6 °C in SSP5-8.5 in continental and northern

Europe. In the Mediterranean region, the decrease of precipitation (-10 to 20% in Spain, Italy and North Africa) accentuates the drying.

*Africa:* The most striking changes are predicted in the Sahara region. Since the initial aridity index is inferior to 0.03 (hyperarid region), very slight changes can have a high impact on the percent change. For example, a 40% wetting is observed in south-east Sahara/oriental Sahel region. This increase could be associated with a warming of the atmosphere, resulting in increased

rainfall intensity during the wet seasons, with flood periods alternating droughts ( He et al. 2023, Palmer et al. 2023).

In the Namib and Kalahari deserts, and more generally in South Africa and Namibia, the aridity index is predicted to decrease from -20% in SSP2-4.5 to more than -40% in SSP5-8.5. This is associated with a -10 to -20% precipitation decrease in Maghreb and Kalahari. In addition, the average temperature increases by up to +3-4 °C (SSP2-4.5) and to 4-5°C (SSP5-8.5).

The west coast of Africa also experiences drying, from – 10% change in the aridity index in SSP2-4.5 to -20% in SSP5-8.5.

The warming is limited to 2 to 4 °C in SSP2-4.5 and 3 to 5°C in SSP5-8.5, but changes in precipitation range from a slight increase in SSP2-4.5 (no more than +10%) to a slight decrease in SSP5-8.5 (less than -10%).

In the Arabic peninsula, the south (Yemen and Oman) is marked by an increase of aridity index, more widely spread in SSP2-4.5 than in SSP5-8.5. A drying in the North of the peninsula with decrease of aridity index of more than 40% in 2070-2100 is noted in Iraq, North of Saudi Arabia, Jordan. In SSP3-7.0, there is no trend of wetting, and a drying is observed mainly in the

centre of Saudi Arabia.

*Asia:* The western part of Asia is already largely composed of drylands, with arid deserts such as the Karakum and Kyzyl-kum deserts. However, the aridity index continues to drop in the short term as well as in the long-term.

In the Indian subcontinent, the aridity index varies between -10 and 10%. The most important changes occur in the South-West, in the province of Kerala. An increase of aridity index between 10 and 20% at the frontier between India and Pakistan,

as well as in Pakistan, in the region of the Thar desert that is currently classified as semi-arid or arid. The changes in the south





can be explained by the temperature increase of 1 to 2°C in SSP 245 over most of the subcontinent. In the SSP5-8.5, this increase reaches up to 3 to 4 °C. This is the only region in India where the precipitation intensity does not change or slightly decreases compared to 1970-2000. In the north-west, the temperature increases much more (2 to 3 °C in SSP2-4.5 and 4 to 5 °C in SSP5-8.5), but precipitations also increases particularly along the border between India and Pakistan (+10-20% in both

SSP). This increase in rainfall therefore counterbalances the increase in temperature.

In the North-East of Russia, the aridity index decreases by - 20 to - 30%, while some small regions have an increasing aridity index. Models predict an increase of precipitation in this region by 10-20% and 20-30% in SSP2-4.5. This increase goes up to 50% in the SSP5-8.5. Temperature will also increase by 4 to 5 °C in SSP2-4.5, 5 to 8°C in SSP3-7.0 and 7 to 9 °C in SSP5-8.5. In SSP5-8.5, the effect of temperature on evapotranspiration dominates, and the drying occurs mainly along the coast from

the Bering Strait to South Korea and Japan.

A large band in the south of China has a decreasing index down to 20% in SSP2-4.5. This corresponds to the currently humid part of China that warms the most (semi-arid and arid regions, in the West which warm much more but do not experience such a decrease in aridity index), while little change in precipitation in any of the 3 SSP. In SSP3-7.0 and 585, this area is much larger.

Some regions of south-east Asia also experience a slight drying (less than 20%). In SSP5-8.5, the areas experiencing drying are the same, but more widespread; while in SSP3-7.0, there are a few changes in aridity across the region except in Thailand, Cambodia and Vietnam. Temperature increase is uniform and limited in this region: 1 to 2 °C in SSP2-4.5, 2 to 3 °C in SSP3-7.0 and 4 to 5 °C in SSP5-8.5. Precipitation changes little in SSP2-4.5 and SSP3-7.0, with changes falling in the range - 10%/+10%. However, up to +30% increase is expected on some islands in SSP5-8.5.

*Oceania:* In SSP2-4.5, a decrease of aridity index by 10 to 20% is observed over almost the entire island of Australia and New-Zealand. There is little change between the periods 2030-2060 and 2070-2100, indicating that the drying will occur in a short term. In SSP5-8.5, the aridity index changes only on the west coast in 2030-2060 and in New Zealand, but decreases much more in 2070-2100, down to -40% in the west and in the south. These areas correspond to the regions in which the annual precipitation decreases between -10 and -20% by 2100.

A decrease in aridity index is projected for central Australia in SSP2-4.5 and SSP5-8.5. In SSP2-4.5, the decrease is between -10 and -20%. Precipitation changes slightly, between – 10 and + 10%, but the temperature increases by 1 to 3 °C. In SSP5-8.5, the aridity index decrease reaches -30%, with almost no change in precipitation except for an increase of up to 40% in the Great Victoria Desert (a region where the aridity index does not change), while the temperature increases between 3 and 5°C. In SSP3-7.0, the temperature increases between 2 and 3°C by 2070-2100, while precipitation increases by up to 40% in central

region of Australia. This results in a decrease of the aridity index on the coastal areas in of the island, but not in the central areas. The period 2030-2060 is actually dryer than the period 2070-2100, with a 10-20% decrease of precipitation intensity resulting in a 10-40% decrease in the aridity index on the island.



Is it worth noting that the SSP3-7.0 is overall different from the others two SSP. The period 2030-2060 is marked by an
increasing aridity index in a large region of Central and Eastern Africa, India and North East Asia. This trend is reversed in
the period 2070-2100, in which the aridity index decreases in most places, in particular in the Arabic peninsula, in India, in
North-East Asia and North-West America for the period 2030-3060. The difference with the two other SSP is less visible for
the period 2070-2100. The humidification observed for the period 2030-2060 could be explained by the introduction in this
scenario of a larger amount of aerosols compared to the 2 others (Cross-Chapter box 1.4, in Chen et al. 2021). This scenario,
presenting more contrasts than the others two, will lead to higher adaptation costs, since the climatic conditions can switch
drastically between the mid-term and the long-term horizons.

Figure 5 shows the average trend followed by the aridity index by continent and by SSP. A visualization by sub-region, as well
as the relative evolution of temperature and precipitation by continent and by subregion, are available in Supplementary (Tables
S2 to S10 and Figs. S10 to S17). The direction and speed of change are clearly visible: for most continents, the most rapid
changes in the aridity index occur between the reference period 1970-2000 and the near future 2030-2060, and continue to
decrease until the end of the century, but at a slower rate.
This is particularly visible for Central and South America, the Mediterranean region and Oceania, in SSP2-4.5 and 585. In
these regions, the changes in aridity index are driven by the conjunction of increasing average temperatures and decreasing
average precipitation. In SSP3-7.0, the average aridity index in Oceania decreases down to -20% in the near future and
increases in the far future.
North America is the continent with the highest internal variability in aridity index, with standard deviation exceeding 100%
of the average value. In SSP2-4.5, the aridity index slightly decreases in the short term, before increasing again in the long
term. In SSP5-8.5, the average aridity index does not change in the near future, but decreases in the long term. In SSP3-7.0,
the drastic increase in aridity index (> +20%), indicating wetter conditions, is followed by a strong decrease (-15%) in the long
term. This responds to the strong wetting of northern part of the continent in the period 2030-2060, followed by the drying of
most of the continent in 2070-2100. In SSP2-4.5 and 585, both temperature and precipitation increase steadily, up to an average
of +7.5°C in 2070-2100 for SSP5-8.5 and a 20% average increase in precipitation over the same period. This is again driven
by the northern part of the continent, which experiences the most drastic changes.
In Europe, the aridity index decreases steadily in the three SSP. In SSP3-7.0, the mid-term decrease is only half of the long-
term decrease, while in SSP2-4.5 most of the decrease happens in the first half of the century. Both temperature and
precipitation increase in SSP2-4.5 and SSP5-8.5, but the influence of the temperature on the potential evapotranspiration
exceeds that of the increase in precipitation, as the aridity index decreases. The strongest decrease is observed for SSP3-7.0 in
the period 2070-2100. In some regions, the near future will bring a wetter average climate (South and East Asia, Oriental
Sahel), but the trend will be reversed by the end of the century. The North American case is the one with the highest
discrepancy, but also with the greatest intermodal variability.







**Figure 5** **Evolution of the mean aridity index in % (up) and of the mean precipitation anomalies (%) and temperature anomalies (°C) (down) by continent in the Socio-economic pathways 2-4.5, 3-7.0 and 5-8.5 (Polar continent, including Greenland, Iceland and Svalbard islands, are not included)**



## 4.2 Towards more drylands

In analysing the changes in aridity index globally it can be seen that significant changes occur even in areas that are considered to be cold or humid. In addition, the change in aridity index sometimes leads to a change in aridity category.

Figure 6 shows the proportion of each aridity category is represented by SSP and by period. The overall proportion of drylands (hyperarid, arid, semi-arid and dry subhumid areas) increases in all SSP but the majority of the land (excluding Antarctica)
remains in the "humid" or "cold" categories. For example, Central America is classified as "humid" in the CMIP6 multimodel average for the reference period 1970-2000, but will experience drastic decreases in aridity index, especially in SSP5-8.5, without leading always to a change in aridity category. This is also noted in South America and Europe (Figure 4).

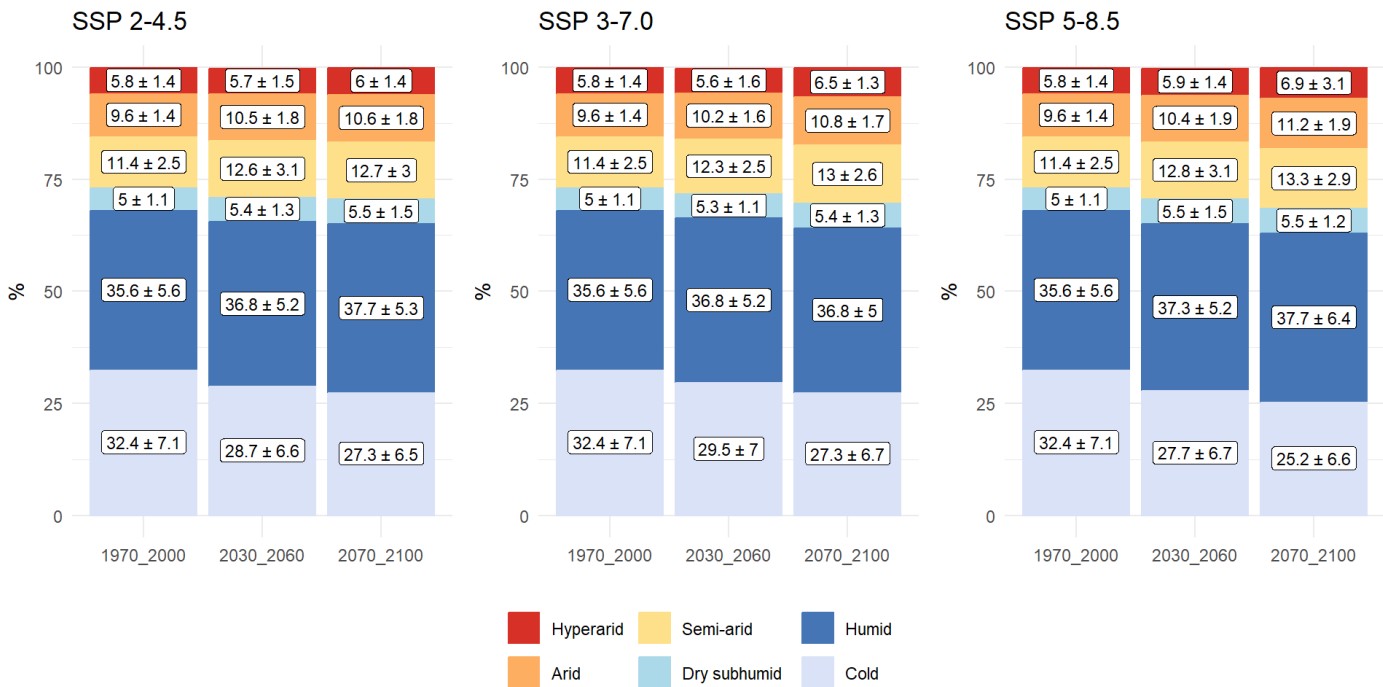

**Figure 6 - Evolution of percentage of aridity categories in the Socio-economic pathways 245, 370 and 585. The proportion of each aridity category represents the average of the 13 CMIP6 models, accompanied by the standard deviation.**

The most extreme projected changes in category, corresponding to the SSP5-8.5, are represented on Figure 7. The corresponding figures for SSP2-4.5 and SSP3-7.0 are available in Supplementary (Figure S8).

*North-America:* Drylands in North America are expanding northwards and southwards. The Sonoran Desert becomes
increasingly arid and the Chihuahuan Desert expands to the south, east and north. A dry subhumid zone appears North-East of the Great Basin Desert, in a region that used to be humid.



*Central-America:* The dry subhumid areas in Cuba, Haiti and the Yucatan Peninsula become semi-arid. The dry subhumid/semi-arid area in the north of Venezuela becomes mostly semi-arid.

*South-America:* The dry subhumid/semi-arid area in Argentina and Paraguay becomes mostly semi-arid and extends towards
Brazil and Bolivia. Some other areas in Patagonia become semi-arid. Finally, a Brazilian region in the east becomes subhumid, when it was classified as humid before.

*Europe:* Little changes, except that the cold to humid limit moves northwards.

*Mediterranean basin:* The hyperarid areas of the Sahara moves northwards, and the semi-arid North African regions become arid. The semi-arid and dry subhumid areas expand in Spain, Italy, Greece and Turkey.

*Africa:* The Arabic Peninsula becomes mostly hyperarid. In South Africa, the semi-arid and arid areas expand around the Namib and Kalahari deserts. In the Horn of Africa, some regions shift to a wetter category. The north-east of Somalia shifts from arid to semi-arid.

*Asia:* Most of the category changes occur in the western part of Asia. The northern boundary of the central Asian deserts (Kara-Kum, Kyzyl-Kum) moves northwards, but no major changes occur in the Taklamakan and Gobi deserts.

*Oceania:* In Australia, the arid and semi-arid areas expand towards the north-east coast, which was previously dry subhumid or humid.



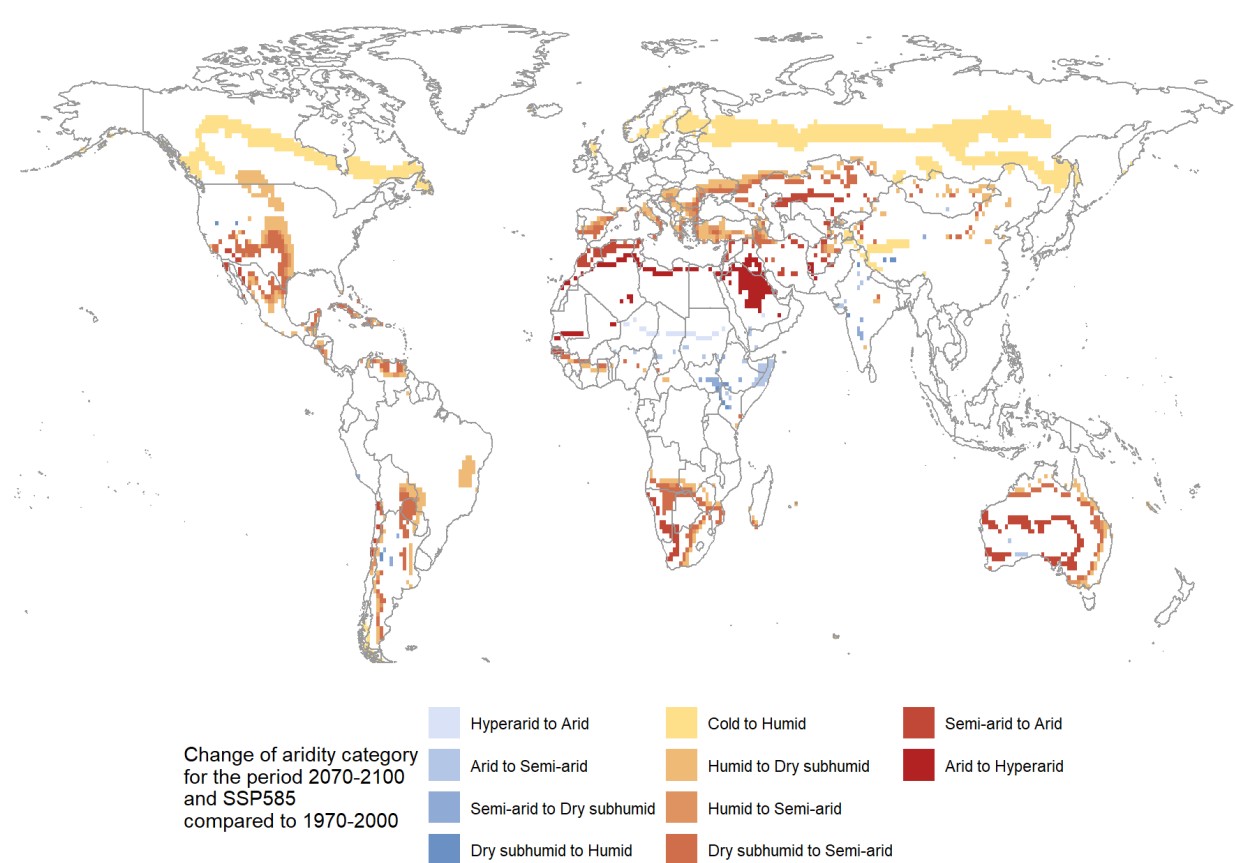

**Figure 7 – Grid cells changing towards a dryer category compared to 1970-2000 for the SSP5-8.5, period 2070-2100**

Overall, we find that with the PM equation on our CMIP6 data, the extent of drylands during the reference period is 31.8%. This extent increased by 3%, 3.9% and 5.1% in SSP 2-4.5, SSP 3-7.0 and SSP 5-8.5, respectively. These trends meet the results presented in the UNCDD report (Vincente-Serrano et al. 2024), with North-America, Latin-America and Europe being the continents most impacted by drying trends. In the report, the initial extent of drylands calculated is higher than in our study (40.6% of the land area for the reference period 1990-2020), but the projected expansion slightly lower (around 2% in SSP 2-

4.5, less than 3% in SSP 3-7.0, around 4% in SSP 5-8.5). To further compare our data with the UNCDD report, we used the Thornthwaite evapotranspiration as a temperature-based method to compute aridity projections with our data. The results are shown in  Table 6. We find that the extent of drylands was lower for the reference period with the Thornthwaite equation (27.2%), but the increase is higher in all SSP compared to the increase calculated with the PM equation: the increase is 3.8% in SSP 2-4.5, and 8.6% in both SSP 3-7.0 and 5-8.5.






These results are much lower than the changes predicted by Huang et al. (2016) with CMIP5 data, using the Penman-Monteith evapotranspiration. The authors calculated a 23% increase of dryland area (reaching 56% of total land area) in RCP8.5, whereas we only have a 5.1% in SSP 5-8.5, and 11% increase (reaching 50% of total land area) in RCP4.5, while this increase is only

of 3% in SSP 2-4.5. These discrepancies can be attributed to the difference between the SSP and the RCP scenarios, and potentially to the better representation of rainfall patterns in CMIP6 compared to CMIP5 (Du et al. 2022).

**Table 6 - Proportion of drylands in % of total land area (multimodel average), using Penman-Monteith and Thornthwaite**
**evapotranspiration, for the 3 studied SSP**

| Period | SSP 2-4.5 | | SSP 3-7.0 | | SSP 5-8.5 | |
|---|---|---|---|---|---|---|
| | Penman-Monteith | Thornthwaite | Penman-Monteith | Thornthwaite | Penman-Monteith | Thornthwaite |
| *1970-2000* | *31.8 ± 6.5* | *27,2 ± 7.7* | *31.8 ± 6.5* | *27,2 ± 7.7* | *31.8 ± 6.5* | *27,2 ± 7.7* |
| 2030-2060 | 31.5 ± 6.4 | 29.6 ± 7.8 | 33.4 ± 6.8 | 30.3 ± 7.7 | 34.6 ± 7.9 | 30.8 ± 7.9 |
| 2070-2100 | 34.8 ± 7.7 | 31.0 ± 5.5 | 35.7 ± 6.8 | 35.8 ± 8.7 | 36.9 ± 9.1 | 35.8 ± 10.9 |





## 5 Conclusion

An ensemble of 13 CMIP6 models was evaluated to compute the aridity index. The multimodel average was evaluated against
two databases, Worldclim and ERA5, which include observations and reanalyses. The CMIP6 multimodel average predicts a
world that is slightly wetter world than observed today, in particular in the North-Eastern Brazil, where the arid area is not
well simulated.

The CMIP6 multimodel average was used to identify future drying and wetting trends in terms of aridity index in the future in
most areas, in three different Socio-Economic Pathways. These scenarios represent different possible futures: in SSP2-4.5,
climate changes are limited and therefore the patterns of change for aridity index and aridity categories are also less visible
than in the SSP5-8.5, which represents the scenario with the largest increase in global temperature. The final scenario, SSP3-
7.0, lies in between but with opposite trends between the near and far futures. As a result, many areas are expected to become
wetter in the mid-term, but drier in the long-term. This is the case in North-America, Africa and Asia where the aridity index
is expected to increase in the mid-term, and then drastically decrease to levels comparable to the other two SSP. This would
result in higher adaptation costs compared to the SSP2-4.5 and SSP5-8.5. In the three scenarios, the Mediterranean basin and
Central America are the regions with the largest decrease in the aridity index. South-America, Europe and Oceania suffer from
significant decreases, but limited to -20%. Overall, a decrease of the aridity index is observed for all continents in the far
future.  Most of the changes already occur for the period 2030-2060 and remain or continue in 2070-2100. Significant changes
of aridity index are also expected within climate zones, in particular in the humid zone, although these changes in the index
are not affected by a change in category. The redistribution of arid areas by the end of the century is similar to today's map,
with an expansion of arid zones towards the periphery of existing zones. Changes to wetter categories are only observed in the
Horn of Africa.

Conclusions for local ecosystems drawn on these results must consider that there is no direct translation between a change of
aridity index and the impact on ecosystems. On the one hand, a main caveat of the Penman-Monteith method is that it assumes
a fixed stomatal resistance. However, with increasing $CO_2$ concentration, this resistance also increases, reducing in turn the
water loss. As a result, evapotranspiration calculated with "historical" resistance value overestimate future evapotranspiration
(Yang et al. 2019). Other ways of estimating evapotranspiration have been suggested, for example by directly using the net
radiation that is a direct product of climate models (Greve et al. 2019) or by introducing a $CO_2$-term in the equation (Lian et
al. 2021), that result in reduced evapotranspiration and therefore less significant drying trends. On the other hand, the aridity
index is a simple proxy that does not allow to discriminate between the drivers of change in a given ecosystem. For example,
Denissen et al. (2022) use an "Ecosystem Limitation Index" that differentiate situations in which the primary production is
limited either by water or energy limitation. The crossing of certain threshold can also be studied, as in Berdugo et al. (2020)
that distinguish three phases in land degradation. Finally, seasonality is not taken into account here, while changes in the length
of the dry and wet seasons could lead to shifts in vegetation even in humid areas (Xu et al. 2022).





**Data availability**

The codes and data used for the statistical analysis are available on Github at https://doi.org/10.5281/zenodo.14230120

**Author contribution**

AD initiated the study, setting the framework and direction of the present work. CC obtained the data, developed the code, performed the data analysis, prepared the figures and redacted the first draft. SA and JB contributed to overcome challenges 560 with the data and to answer conceptual questioning during the data analysis. All authors reviewed the manuscript and contributed to its readability.

**Competing interests**

The authors declare that they have no conflict of interest.

**Funding**

The research leading to this study was co-funded by IRD (Institut de Recherche pour le Développement; France) grant number "UMR IGE Imputation 252RA5" and by RNER-CC (AFD-C2D) project implemented in the CNCCI (Côte d'Ivoire National Center of High Performance Computing).



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
