# Peer review of "Global projections of aridity index for mid and long-term future based on CMIP6 scenarios"

_EGUsphere, 2024_

## Author Comment (AC1)

This study provides a well-structured and comprehensive analysis of global aridity projections based on CMIP6 scenarios. The results are presented clearly, and the methodological approach appears to be sound and well explained. The study is relevant for understanding long-term trends in desertification and future climate impacts.

**Comments:**

- Aridity classification - The manuscript primarily focuses on desertification, but only includes 1–2 humid categories. Would it be possible to shift the focus slightly toward transitions between different aridity index (AI) classification states rather than focusing exclusively on desertification? If the authors prefer to maintain the current classification, a justification for this choice would be helpful.

Thank you for this comment. The article studies processes of drying and wetting in terms of aridity index, not only comparing the classification in "aridity categories". The classification we used is the same as the one reported in UNESCO 1979.

The aridity index is meaningful only when the ratio precipitation / evapotranspiration is inferior to 1. The "drylands" categories are defined up to a ratio AI = 0.75, that get close to this equilibrium. It does not make sense to define more "humid" categories in the study when studying the shift towards dryer conditions. In addition, that would prevent our categories to be compared with previous studies.

- The AI classification used in this study appears to be slightly different from the classification used by the IPCC Sixth Assessment Report and UNCCD (2024), also cited in this study. See: [Dry sub-humid ($0.5 \leq AI < 0.65$), Semi-arid ($0.2 \leq AI < 0.5$), Arid ($0.05 \leq AI < 0.2$), Hyper-arid ($AI < 0.05$)]. It is only a minor change to the classification but it would make it easier to compare your assessment to more recent publications. See for reference: e.g. Figure CCP3.1 in Mirzabaev, A., L.C. Stringer, T.A. Benjaminsen, P. Gonzalez, R. Harris, M. Jafari, N. Stevens, C.M. Tirado, and S. Zakieldeen, 2022: Cross-Chapter Paper 3: Deserts, Semiarid Areas and Desertification. In: Climate Change 2022: Impacts, Adaptation and Vulnerability. Contribution of Working Group II to the Sixth Assessment Report of the Intergovernmental Panel on Climate Change [H.-O. Pörtner, D.C. Roberts, M. Tignor, E.S. Poloczanska, K. Mintenbeck, A. Alegría, M. Craig, S. Langsdorf, S. Löschke, V. Möller, A. Okem, B. Rama (eds.)]. Cambridge University Press, Cambridge, UK and New York, NY, USA, pp. 2195–2231, doi:10.1017/9781009325844.020.

In our study, we use the threshold for aridity categories defined in the UNESCO aridity map of 1979. This map was established using the aridity index calculated with the Penman-Monteith evapotranspiration equation, where the thresholds were defined according to the major bioclimatic categories. Later, the Cross-Chapter Paper 3 and the UNCCD report use slightly different boundaries for aridity categories based on the categories adopted by the World Atlas of Desertification published in 1997 (second edition). In the Atlas, the evapotranspiration is calculated using the Thornthwaite equation. The difference of thresholds compared to UNESCO 1979 is deliberately made because of the tendency of the Thornthwaite equation to underestimate evapotranspiration in dry zones, and overestimate it in wet zones.

This article uses the Penman-Monteith evapotranspiration, therefore we kept to the thresholds defined for this equation. The resulting categories are comparable with the categories of the

Cross-Chapter and UNCCD report, that both use thresholds adapted to the use of Hargreaves and Thornthwaite equations.

A paragraph explaining why the thresholds are different has been added line 113:

*The climate is then classified into 5 classes depending on their aridity index. The thresholds used in this article were defined in the explicative note of the UNESCO (UNESCO 1979) on the map of the world's arid regions, based on the bioclimatic characteristics of these areas. These thresholds are slightly different to those used in the UNCCD report on desertification (Vincente-Serrano et al. 2024) and in the dedicated chapter of the IPCC AR6 (Mirzabaev et al., 2022), because these two reports use respectively the Hargreaves and Thornthwaite evapotranspiration equations. The two equations underestimate evapotranspiration in dry areas and overestimate it in humid areas. In their case, the hyperarid areas are defined by an aridity index inferior to 0.05 (instead of 0.03 here) and humid areas with an aridity index superior to 0.65 (instead of 0.75 here) in order to match the categories defined with the Penman-Monteith equation.*

- Will the dataset produced in this study be made publicly available? A dataset of time-series AI classifications would enable further studies on system-state transitions, which could be valuable for assessing long-term desertification and land degradation trends. Making such data accessible would enhance the impact and usability of this research.

Thank you for noticing this. The dataset was supposed to be publicly available on the Zenodo repository associated with this work. The dataset will be added in the next release.

- **Formatting comments:**
  - There are some inconsistencies in citation formatting. For example, "et" appears to be used as "and" in some cases (e.g., lines 56, 59, and 73). Standardizing the citation format to English would avoid confusion.

This has been corrected for the next submission.

  - Lines 99–100: Please check the units—there appears to be a discrepancy of three orders of magnitude between mJ and MJ.

This has been corrected for the next submission.

  - Lines 545 and 548: The formatting of "$CO_2$" should be corrected.

All occurrences have been corrected for the next submission.

  - Table 2 should be formatted for easier readability.

Table 2 has been modified to make it more easily readable.

---

## Author Comment (AC2)

Review comments on "Global projections of aridity index for mid and long-term future based on CMIP6 scenarios by Crapart et al.

The authors evaluated AI and dryland distribution projected by 13 CMIP6 models for three different socio-economic scenarios and for two timeframes of 2030-2060 and 2070-2100. The evaluation was done against WorldClim and ERA5 datasets for 1970-2000. Their projections indicate significant (SSP2-4.5 and SSP5-8.5) or limited (SSP3-7.0(drying) for mid-term but more consistent and continuing drying in the long-term.

Systematic analysis of aridity in the mid-term and long-term future periods for a selection of representative emission scenarios is well within the interests of HESS readership and the use of 13-member ensemble of CMIP6 outputs add values to the future aridity and drought studies. However, the manuscript needs clarifications in some of the methods and results presented, justification or change of the ensemble CMIP6 sampling, and overall improvement in writing before it is considered for acceptance to HESS. I recommend a major revision. More detailed comments are provided below.

General comments

The manuscript contains analysis of internal variability in CMIP6 members (Section 3.1) and evaluation of CMIP6 against ERA5 and WorldClim (Section 3.2) over a historical period of 1970-2000. An important utility of CMIP6 evaluation over historical period is that it can provide insights on the behavioral features of projections produced by each ensemble member in terms of bias, trends, dynamic ranges (seasonal or interannual), mean difference, etc. Analysis of projected AIs needs elaboration based on the results of Sections 3.1 and 3.2.

**Thank you for this comment. Firstly, I need to highlight a mistake that was made in the first version of the manuscript. The term "internal variability" was unproperly used. We intended to handle the topic of dispersion of the ensemble rather than the natural variability of climate. This terminology will be clarified in the revised version.**

**The section 3.1 and 3.2 allow us to evaluate the areas in which CMIP6 historical models compare the less with reanalysis, in order to assess the similarity between the multimodel mean of CMIP6 historical runs, and Worldclim/ERA5. Using this section to assess the relevance of the projections, as is suggested in the next comment, would definitely add value to the results. The sections 4.1 and 4.2 will be improved in this regard in the revised manuscript.**

In addition, for CMIP6 based projections, the candidate models are often 'conditioned' by comparing their results with historical observation or reanalysis data. The conditioning can be based on different evaluation metrics depending on the information of interests (e.g., trends vs. absolute values) for the projected period. The authors are recommended

to try conditioning of the ensemble to improve the reliability of the ensemble of projected AIs. Assessment of the Section 3.2. appears to be an excellent source of information for this purpose.

**Thank you for noting this. I understand this comment as related to the previous one. In section 3.2, we compare the aridity index and classes obtained for historical CMIP6 multimodel mean, to the aridity indices obtained with Worldclim and ERA5. We can therefore determine which areas are most susceptible to deviate from "actual" results. For example, a region that is more humid in CMIP6 than in ERA5 or WDC is also likely to be more humid in CMIP6 projections. This is not sufficiently done in the current version of the manuscript, and will be included in the final version.**

Results include evolving aridity, for some regions, from wetting or moderate drying for 2030-2060 to more consistent or stronger drying for 2070-2100, particularly for SSP3-7.0. Given that the projections are derived from models, the manuscript may include more process-based explanation for the inconsistent changes between the mid-term and long-term projections.

**This is indeed an interesting point to investigate further. As mentioned in our manuscript, the main difference between SSP 3- 7.0 and the two others is that this SSP was designed to include low or no policies on air quality and emissions. The impact of aerosols on radiative forcing, as well as on regional precipitation or temperature, is complex. The paragraph focusing on the particularities of this SSP (lines 421-429) will be enriched by more details about this scenario and how aerosols can have different impact locally, using for example the work of Lund, Myhre, and Samset (2019) and Collins and al. (2017) who studied the impact of aerosol emissions in CMIP6 projections.**

It appears that there exists similarity between the manuscript and the UNCCD report (Vincente-Serrao et al., 2024) cited in the manuscript. Provide the difference between the manuscript and the UNCCD report (Vincente-Serrano et al., 2024) in methods and datasets. The UNCCD report appears to include comprehensive assessment of similar projections with minor differences in the projected time windows.

**There are indeed similarities between these two works, as this paper is intended to support the efforts of the UNCCD in investigating future desertification trends. However, there are noticeable differences that make these two studies complementary.**

**The first and main one is the use of the Penman-Monteith formula for potential evapotranspiration in our article, while the results in the UNCDD report are based on the Hargreaves-Samani formula. The Hargreaves-Samani formula is based on minimum and maximum temperature, hence does not take into account the**

projections in wind speed, surface radiation etc. Therefore, we expect the Penman-Monteith equation to give more accurate results for aridity index projections.

There are also a couple of methodological differences between our two studies. For example, the UNCDD report uses projections for 6 GCM (we use 13) and historical runs from 10 GCM (where we use the same 13 GCM as for the projections). In addition, the "cold" category in the UNCDD report is defined as areas where the mean annual temperature is <10°C, following a Köppen-Geiger definition, while we use the same approach as in the last World Atlas for Desertification: annual PET < 400 mm.

Since the raw data of the UNCDD report is not available, it is difficult to compare exactly our results with theirs. However, we compared the evolution of dryland proportion by continent in their report and in ours. A paragraph will be added in the revised manuscript. Globally, the projections of aridity index with Penman-Monteith results in a higher proportion of drylands in the future compared to the UNCDD report; but there are differences depending on the continent.

Following a comment from reviewer 1, we also added a section in which we compare the proportion of drylands in SSP 2-4.5, 3-7.0 and 5-8.5 if using the Thornthwaite potential evapotranspiration equation, that also relies only on temperature. The proportion of drylands is systematically lower with Thornthwaite compared to Penman-Monteith.

Some paragraphs in Introduction are loosely related to the main focus of the manuscript (e.g., history of climate zones). Tighten words and improve focus in the Introduction section. There are mentions of 30-year or 30 year, but the three periods of analysis are all 31 years (1970-2000, 2030-2060, and 2070-2100), right? Also, description of methods needs improvement for clarity. For example, more detailed description is needed for how results in Figure 1 and Table 4 are produced. Were all 31-year (not 30) data over all grid cells put together for Figure 1? Lines 245-246 indicated a use of 30-year (31?) average, but was it used for all results or just Figure 2?

Thank you for noticing these details.

The introduction will be shortened in the final manuscript.

The periods used for computing projections are all 30 years, for ex 1970-1999. "2000" was used by convenience, but this will be corrected.

In the section 3.2 (Figure 1 and Table 4), the objective is to compare Worldclim, ERA5 and historical runs of CMIP6. The violin plots of figure 1 help displaying the distribution of the values of each parameter for the 3 datasets; and table 4 presents pairwise computations (ERA5 against Worldclim, CMIP6 against REA5 etc) of r2 for

each variable. Details about violin plots and what they represent, as well as the correlation coefficients, will be added.

Specific comments

- Lines 20-21: The subject for the second part the sentence is different from the earlier part. Revise the sentence.

**The sentence will be revised in the corrected ms.**

- Line 21: 'population' would be a more suitable word since 'inhabitants' can refer to the entire animal, unless that is the intended meaning.

**This has been modified.**

- Line 24: Change 'contrarily' to 'in contrast'.

**This has been modified.**

- Line 26: "widespread and persistent".

**This has been modified.**

- Lines 66-67: What does "changes" here refer to? Changes from one category to another over time?

**It refers to the evapotranspiration and aridity index changes. This has been specified.**

- Line 69: Revise the following to a correct expression: "the future downscaled CMIP6 models".

**The sentence has been changed to "to the few variables available in the CMIP6 projections gathered in Worldclim". The meaning of the sentence is that Worldclim offers gridded data from CMIP6 that has been downscaled to resolutions inferior to 100x100 km, for a selection of variables that do not include all the necessary variables for calculating Penman-Monteith.**

- Eq 1 and throughout the manuscript: ET0 has been used to indicate the potential ET in the manuscript but it is commonly used for the reference ET. 'PET' would be a better choice to mean the potential ET.

**We used ET0 throughout the manuscript because it is the way it is defined in Allen et al as the reference evapotranspiration, based on the Penman-Monteith method. Thus, we used it as a synonym of PET in this manuscript. For clarification and to be consistent with other papers on the topic, the notation will be changed to PET.**

- '2-m' instead of '2m' when it is used as adjective.

**This has been done, in particular in table 2.**

- Table 1 and throughout the manuscript: AI = 0.75 is used as threshold between 'Dry subhumid' and 'Humid'. UNESCO (1979) uses AI = 0.65 https://www.ipcc.ch/report/ar6/wg2/figures/chapter-ccp3/figure-ccp3-001

**This can indeed be confusing. This point is answered below.**

- Lines 120-121: Rewrite this sentence for clarity.

**The point here is precisely to explain why a threshold of 0.75 is used with the Penman-Monteith equation, contrarily to the 0.65 threshold used with the Hargreaves and Thornthwaite equations (as in UNESCO 1979). This is due to the constatation that Hargreaves and Thornthwaite equations, also commonly used to compute the potential evapotranspiration, tend to overestimate evapotranspiration in humid areas. Therefore, the threshold for "humid" areas is lower when using Hargreaves or Thornthwaite, compared to using Penman-Monteith.**

- Line 120: Use 'lower' or 'smaller' in place of "inferior to".

**This has been modified.**

- Line 129: What does the "3 datasets" refer to?

**The 3 datasets refer toERA5, Worldclim and CMIP6. This has been specified.**

- Section 2.2.1: Use plain paragraphs or summary of lists in tables instead of bullet points.

**The bullet points were replaced by plain paragraphs.**

- Table 1: Highlight the quantities that exhibit large deviations from the central tendency (mean or median).

**I assume that this comment holds for table 3? I have formatted the table so that the deviations appear clearly.**

- Table 2 and throughout the manuscript: Units appears to be inconsistent or not in compliant with HESS style.

**The HESS guidelines recommend to use SI units, expressed with negative exponents. These are the units used in the manuscript are SI units, but we expressed them with separative dots, which we will remove in the final manuscript consistently with the recommendations of the International Bureau of Weights and Measure. We also corrected inconsistencies in units writing in diverse parts of the manuscript.**

**Some differences will remain when necessary. For example, Table 2 presents the variables and their units as available in Worldclim, ERA5 and CMIP6. They are not always consistent: for example, precipitation in Worldclim are expressed in mm $y^{-1}$, while they are available in kg $m^{-2}$ $s^{-1}$ in ERA5 and CMIP6. The variables are all converted to the units corresponding to the Penman-Monteith equation before use.**

**For example, in Figure 1, the variables are presented in the units in which they are inserted in the equation (see equation 2) and available as such in the data that we release with this article.**

- Quality of Table 3 needs improvement (e.g., margins, line thickness).

**Table 3 has been updated.**

- Figure 2 and relevant section: The reference data, ERA5 and WorldClim, feature noticeable discrepancy in AI. What are the sources of them when they both incorporate ground data?

**The information on the provenance of the data is given in section 2.2.1. The Worldclim dataset is a combination of several sources of observation and reanalysis. More details will be added in the final manuscript.**

**One of the main reasons for which there is a discrepancy in AI is the way that surface radiation is computed with Worldclim data (based on latitude only). This impacts in particular the high latitude regions. In addition, in Wordclim, the gridded data (in particular precipitation) is obtained from interpolating observations, which might result in more uncertainties for regions where observations are scarce.**

**References:**

Collins, William J., Jean-François Lamarque, Michael Schulz, Olivier Boucher, Veronika Eyring, Michaela I. Hegglin, Amanda Maycock, et al. 2017. « AerChemMIP: Quantifying the Effects of Chemistry and Aerosols in CMIP6 ». *Geoscientific Model Development* 10 (2): 585-607. https://doi.org/10.5194/gmd-10-585-2017.

Lund, Marianne T., Gunnar Myhre, et Bjørn H. Samset. 2019. « Anthropogenic Aerosol Forcing under the Shared Socioeconomic Pathways ». *Atmospheric Chemistry and Physics* 19 (22): 13827-39. https://doi.org/10.5194/acp-19-13827-2019.